# Signal processing to denoise and retrieve water vapor from multi-pulse-length lidar data

Matthew Hayman[1], Robert A. Stillwell[1], Adam Karboski[1], and Scott M. Spuler[1]

[1]Earth Observing Laboratory, NSF National Center for Atmospheric Research, Boulder, CO, USA

**Correspondence:** Matthew Hayman (mhayman@ucar.edu)

**Abstract.** Recent hardware developments for the MicroPulse DIAL enable the transmitter to switch between output pulses that are "longer" (higher pulse energy) and "shorter" (low pulse energy) in duration on a shot-to-shot basis. While the longer laser pulses broadly result in higher signal-to-noise ratio, they have the shortcoming of blanking the detector in the lowest ranges and smearing out the scene in range. Conversely, shorter pulses enable observations closer to the instrument, smear the scene relatively little, but have low signal-to-noise ratio. In this work, we show that leveraging Poisson Total Variation with forward modeling enables merged estimates of backscatter and water vapor. This signal processing technique leverages the advantages of each pulse length configuration, providing better data availability and higher resolution over a broader altitude range than data processed using only one of the pulse lengths. An intercomparison with radiosondes demonstrates that this new hardware configuration and processing approach enable retrievals of absolute humidity starting at 100 m extending up to 6 km, capturing complex water vapor structure throughout this range. The retrievals are also contrasted with ERA5 reanalysis which suggests that there are instances where the model and reanalysis products are unlikely to produce accurate representation of water vapor fields in the atmosphere, thus emphasizing the value of continuous, high-vertical-resolution active thermodynamic profiling observations.

## 1 Introduction

The MicroPulse DIAL (MPD) is a lidar instrument designed around the concept of multi-instrument network deployment (Nehrir et al., 2011; Repasky et al., 2013; Spuler et al., 2015, 2021). It leverages low-power, high-reliability semiconductor lasers that make it well suited for unattended operation. The combination of the lasers' narrow linewidth – enabling narrow band filtering and interrogation of specific absorption features – with an opto-mechanically stable coaxial transceiver architecture – enabling a narrow receiver field-of-view ($\approx 100 \mu rad$) – allows the MPD to provide quantitative measurements of water vapor (water vapor DIAL technique), temperature (oxygen DIAL technique) (Hayman et al., 2024b; Stillwell et al., 2020) and aerosol backscatter coefficient (high spectral resolution lidar) (Hayman and Spuler, 2017) over an altitude range of 300 m to 6 km, during day and night.

The MPD is unique in that it's transmitter is entirely based on low-power diode lasers but extends to robust quantitative data products well beyond the qualitative underconstrained retrievals of backscatter lidar. MPD processing provides quantitative data products directly linked to atmospheric and model state parameters more commonly associated with high performance systems using more complex transmitter architectures. Unlike solid state crystal lasers, the laser pulse length of the MPD's semiconductor transmitter can be easily modified, setting up an opportunity to tune the parameter. Longer pulses increase the average transmitted power, allowing for a higher signal-to-noise ratio (SNR), better long-range performance, and reduced integration times, but they also reduce range resolution, by smearing out the atmospheric scene. In addition, when used with a coaxial transceiver such as in the MPD design, the lidar receiver is effectively blinded by the outgoing laser pulse (with some additional recovery time) and therefore the pulse length restricts the minimum altitude of the vertically pointing instrument. This sets up an inherent trade-off between higher altitude and shorter time-resolution observations (requiring longer, higher energy pulses) and observing near the surface (requiring shorter pulses). For the standard MPD architecture, a $1\mu s$ transmit pulse results in an absolute minimum capture altitude of 150 m, but additional recovery time effects from the exiting pulse extend this actual minimum altitude higher (300 - 500 m depending on instrument).

In the past, we have set the MPD laser pulse length to balance between the trade-offs of SNR, minimum altitude, and range resolution (generally setting to 1 $\mu s$ as an acceptable balance in the trade space — we should note that the plethora of scientific questions and lack of robust analysis on the observational requirements thereof for instrumentation make any such trade a fairly flimsy analysis). The need to profile close to the ground, in particular, has driven us to consider how to better leverage the flexibility of the diode laser transmitter. In this approach, we have developed a timing and control unit that allows us to program the laser pulse pattern of the MPD (Stillwell et al., 2025). In our initial testing of this hardware, we operated the water vapor DIAL channels of the MPD in a mode where the lasers alternated between transmitting long ($1\mu s$) and short ($200ns$ or $100ns$) laser pulses. The concept behind this development is that data collected using long pulses would enable observations at long ranges and other signal-starved regimes, while the short pulses would enable observations closer to the surface and reduce range smearing.

In Stillwell et al. (2025) the hardware development of this concept is described and the potential of the approach is demonstrated using a conventional water vapor DIAL retrieval (direct inversion with the DIAL equation). As such, the inversion of water vapor and backscatter fields is over-constrained (two unknowns – absolute humidity and attenuated backscatter cross section[1] – with four observations – online and offline wavelengths at two different pulse lengths). In this configuration, applying the DIAL equation produces two separate estimates of absolute humidity (one for each pulse length); which must be subsequently merged to produce a unified water vapor estimate. This manual merging process has some key drawbacks recognized in Stillwell et al. (2025) in that the artifacts can appear in the merge region between the independently estimated products (see Figure 5 of Stillwell et al. (2025)), information content is discarded and therefore retrieval precision is not as high as it could be, and the potential benefits of short pulses enabling recovery of high resolution features (more apparent in cloud/aerosol lidar) are lost. In addition, leveraging joint retrievals with over constrained systems tends to act as a useful quality control

---

[1]The forward model employed here includes three variables, but the only channel gain is broadcast across all range dimensions. For any one pixel one might say there are slightly more than two unknowns but less than three.

check on the recovered data products because the inability to fit all data sets indicates inaccuracies or invalid assumptions in the instrument model used for the inversion algorithm. Finally, in addition to low altitude observations, short pulse data tends to be valid nearer cloud edges than long pulse. Manually adjusting weighing functions near clouds to leverage this benefit would be a fairly complex task as the transition regions would likely be quite variable. This makes it impractical to merge long and short pulse data in and around clouds and instead accept lower data availability.

In this work, we demonstrate how advanced processing methods like Poisson Total Variation (PTV), which use forward models, are well suited to the problem of retrieving parameters that are over-constrained and can provide a high quality estimate of products by seamlessly integrating all of the observation channels. This is done in a way that balances the uncertainties of each observation so that the data product is most responsive to the observations that have the lowest uncertainty based on the detector noise model. It also provides continuity in the estimated fields where data from one pulse configuration may not be accurate or available (e.g. low altitudes and near cloud edges).

PTV was originally introduced to the atmospheric lidar field in Marais et al. (2016), where it was shown to provide unprecedented denoising capability for High Spectral Resolution Lidar (HSRL) retrievals of backscatter coefficient, lidar ratio, and extinction coefficient. That technique was then adapted to retrieve water vapor from MPD observations (Marais and Hayman, 2022), and shown to improve both data quality and operational range of the instrument. The denoising capability of PTV was further demonstrated in Hayman et al. (2024a) where it was shown that extremely high-resolution time-correlated single-photon counting (TCSPC) data produced by the same low-power diode lasers could be processed using the technique and enables backscatter observations of structures at 0.02 second and 0.75 m resolution. Finally, the ability of PTV to denoise and account for complex cross-dependencies in multi-parameter observations was shown in Hayman et al. (2024b) where the temperature, absolute humidity and backscatter ratio were simultaneously retrieved from all six of the MPD observation channels comprising a combination of water vapor DIAL, oxygen DIAL, and HSRL architectures, enabling an entirely standalone thermodynamic profiler (without assumptions or external data required).

Where in previous works we have focused on retrieving data products with varying dependencies between channels, in this work, we establish the potential for PTV to robustly solve over-constrained retrieval problems and leverage advances in hardware uniquely enabled by diode-laser-based lidar. The retrieval presented here aims to leverage information content from both long- and short-pulse data to better constrain backscatter and water vapor observations starting at 100 m and extending up to 6 km. We roughly estimate the effective resolution of MPD water vapor retrievals are about 100 m (not to be confused with a capture resolution of 7.5 m), thus suggesting the MPD is effectively observing water vapor to the surface.

While deconvolving the laser pulse has always been a component of PTV processing for MPD data, the multi-pulse-length nature of this problem further leverages that capability through the hardware development that enables capture of data to better constrain the inversion problem. We show here that by employing PTV with a forward model, MPD retrievals have lower error over a larger altitude range than long or short pulse data alone. This is demonstrated on both backscatter and water vapor data, where the water vapor retrievals are validated against collocated radiosondes. While this work focuses on water vapor retrievals, there is also sufficient demonstration that this hardware/processing approach would likely have benefits aerosol and cloud lidars (e.g. HSRL, Raman) and may have important implications for airborne and space borne platforms. In addition, the

concept of capturing a single data product from multiple related observations has potential beyond multi-pulse observations (e.g. WV DIAL using more than 2 wavelengths).

## 2  Methods

### 2.1  Poisson Total Variation

PTV is a signal processing approach that simultaneously inverts and denoises photon limited observations using a regularized maximum-likelihood fitting approach. Estimated variables are forward modeled and projected onto observed photon counts then subsequently numerically optimized to obtain the best fit while reducing sensitivity to noise.

Noise suppression is implemented through total variation regularization (a penalty in the objective function for changing the estimated variable) which effectively fixes the solution basis set to be piece-wise-constant functions. The ultimate effect is that PTV adaptively average across regions of correlated signal, to produce an optimally averaged data product that has low sensitivity to noise. Under standard processing conditions, this optimal averaging is not heuristically determined but rather verified against holdout data obtained by splitting half the signal into validation data. The result is a retrieved data product in which averaging occurs across patches of correlated structure so that the resolution varies across the 2D image and is not constant across the time or range axis.

The general water vapor DIAL signal model used here is similar to that employed in (Marais and Hayman, 2022) where now the laser pulse may be varied to create different channels for combinations of transmitted wavelength and pulse width. The mean photon flux incident on the detector for a given pulse length and wavelength is forward modeled from estimated variables using the standard lidar equation for DIAL convolved with the transmitted laser pulse.

$$\rho_{c,\lambda}(r) = l_c(r) * \frac{1}{r^2} \tilde{\eta}_c \tilde{\phi}(r) \exp\left(-2 \int_0^r \sigma_\lambda(z) \tilde{n}_{wv}(z) dz\right) + \rho_b \tag{1}$$

where $l_c(r)$ is the laser pulse width of channel configuration $c$, $*$ denotes a convolution in range, $r$ is the range from the lidar instrument, $\tilde{\eta}_c$ is the channel gain (in practice set to 1 for the offline channels), $\tilde{\phi}(r)$ is the backscatter flux common to both online and offline wavelengths (low wavelength sensitivity), $\sigma_\lambda(z)$ is the water vapor absorption cross section at transmitter wavelength $\lambda$, $\tilde{n}_{wv}(z)$ is the water vapor number density and $\rho_b$ is the background flux estimated by averaging the counts at the end of the lidar profile before another laser pulse is fired. Note that because the background is a function of receiver behavior and the same detector is used to acquire optical signals from all four channels, it should be channel independent in the WV DIAL. All estimated variables in the processing are denoted with a tilde ($\sim$) over them.

Since the MPD architecture under consideration here transmits two laser pulses ($c \in \{1\mu s, 0.2\mu s\}$) and two wavelengths ($\lambda \in \{\lambda_{on}, \lambda_{off}\}$) there are more observations than unknowns in a scene. However, the accuracy and precision of the channels will be different depending on the amount of smearing imposed by the laser pulse, the amount of backscatter signal and whether (and for how long) the instrument is blanked by the outgoing laser pulse. As an algebraic inversion, estimating water vapor

from all four channels is not possible, but because lidar observations are inherently noisy, there is a clear need to use as much information as possible to recover a high resolution, low noise data product.

A natural way to solve this problem is to use a forward modeling approach as employed with PTV. This approach estimates the desired parameters by finding estimated variables that optimally project onto all the noisy photon count observations. In this way, one retrieval is produced for each estimated variable, delivering an inherently merged data product. The inherent trade-off in prioritizing the match between different channels is handled through a maximum-likelihood estimation approach, where a noise model for the photon detection process favors better fits to areas of low uncertainty over areas of high uncertainty. Missing data, for example, resulting from proximity to clouds or blanking near the ground, are also handled naturally in this way.

One benefit of over-constrained retrievals is that they can identify weaknesses in the assumed forward models. For example, we have seen instances where retrievals cannot simultaneously fit all channels and therefore indicated errors in the assumption of lidar operation and model. While this outcome can be challenging to address, it ultimately results in better data product fidelity and increased confidence that when the retrieval converges for all channels, the forward model and detection statistical model (both of which represent inherent assumptions in all remote sensing retrievals) are valid for the scene under consideration.

In the estimation process, only the channel efficiency term, $\tilde{\eta}_c$, for the online channel is estimated. This accounts for a scalar differential in the online and offline channels efficiency or transmit power. The offline channel is set to 1 and any necessary scaling to fit backscatter data is absorbed in the backscatter term $\tilde{\phi}(r)$.

In order to estimate the unknowns in the lidar forward model, we minimize an objective function that consists of a maximum likelihood loss function $\mathcal{L}(\tilde{\rho})$ and total variation regularization terms for each estimated variable.

$$\mathcal{O}(\tilde{\phi}, \tilde{n}_{wv}, \tilde{n}_{wv,0}) = \alpha_\phi ||\tilde{\phi}||_{TV} + \alpha_{n_{wv}} ||\tilde{n}_{wv}||_{TV} + \alpha_{\eta_{on}} ||\tilde{n}_{wv,0}||_{TV} + \mathcal{L}(\tilde{\rho}) \tag{2}$$

The loss function leverages the noise model for a non-paralyzable photon counting detector with a known dead-time derived and validated in (Kirchhoff et al., 2025) and given for this application by

$$\mathcal{L}(\tilde{\rho}) = \sum_{c,\lambda,i} w_{c,\lambda,i} \left[ \tilde{\rho}_{c,\lambda,i}(\tilde{\phi}, \tilde{n}_{wv}, \tilde{\eta}_{on}) z_{c,\lambda,i} - y_{c,\lambda,i} \ln \rho_{c,\lambda,i}(\tilde{\phi}, \tilde{n}_{wv}, \tilde{\eta}_{on}) \right] \tag{3}$$

where $w_{c,\lambda,i}$ is a weight applied to the data point which is used to mask invalid data, $z_{c,i}$ represents the detector active time histogram that accounts for dead-time in the detector (Kirchhoff et al., 2025), $y_{c,i}$ are the photon counts for laser pulse $c$ and range bin $i$ and $\tilde{\rho}_{c,\lambda,i}$ is the forward model described by Eq. (1) which relates the estimated data products to the incident photon flux. In this way, the maximum likelihood estimate is able to account for dead-time when the photon flux is relatively constant over the accumulation interval.

The total variation penalties in Eq. (2) force estimated variables to be piecewise constant and suppress fitting to noise. The amount of total variation regularization is determined by the scalar multiplier $\alpha_x$ (for the estimated variable $x$), which is not assumed to be identical for all estimated variables. In order to determine these optimal regularization parameters, we typically employ a thinning process where every other recorded 2 second profile and the remaining profiles are used for holdout

cross-validation of the solution results. Those solutions which optimally fit to the statistically independent validation data are assumed to represent the best balance between bias (over averaging) and noise (under averaging). The thinning process can also be performed randomly (as is used in the masking scheme described below). We do not employ Poisson thinning as has been done in some previous works because observations are not necessarily Poisson distributed, particularly in higher photon flux scenes.

The objective function is minimized using Sparse Poisson Intensity Reconstruction Algorithm (SPIRAL), SPIRAL-TV Harmany et al. (2012); Oh et al. (2013) with the Fast iterative shrinkage algorithm used to calculate gradients Beck and Teboulle (2009). Our Python implementation of this algorithm, based on the PyTorch library, is publicly available for use at https://github.com/NCAR/SpiralTorch with a custom CUDA kernel for accelerated GPU implementation. We have previously detailed the process for minimizing the objective function in Hayman et al. (2024a).

## 165 2.2 Masking

It is common practice to mask cloud structure in DIAL estimates due to the frequent occurrence of biases in retrieved water vapor concentration. These biases are often easy to identify because the retrieved quantities are non-physical (e.g. negative water vapor or unrealistically large values) in the cloud structures. There is relatively little published research on the root cause of data quality issues for DIAL in cloud, and the dominant sources of issues may vary between hardware architectures.

Differences in online and offline sample volumes, laser pulse length convolution, detector nonlinearity, and Rayleigh-Doppler effect are common explanations for the error in most clouds; however, our own efforts to demonstrate and isolate the impact of these and other errors have been difficult to comprehensively validate experimentally. It seems entirely possible that there are additional contributing factors. It is also likely that there are some conditions where DIAL retrievals are valid in certain types of clouds and under certain conditions. For example, Groß et al. (2014) showed DIAL retrievals in high altitude cirrus that

agreed well with an in situ sensor on a second aircraft over a 30 minute period. However, the analysis presented in that work was fairly limited, making it difficult to generalize even to other cirrus cloud cases. For now, it is not entirely clear what all the interacting conditions are that influence DIAL retrieval accuracy or if it is possible to know if those conditions are satisfied from the lidar data alone (meaning no external validation source would be required). For this reason it is often generalized that DIAL data is valid in clear air and most aerosol loaded conditions, but not in clouds.

Our masking routine developed here is based on the hypothesis that much of the bias seen in DIAL observations of clouds originates from averaging over heterogeneous backscatter structure (Hayman et al., 2023)[2]. However, based on ultra-high-resolution observations of clouds by MPD (similar to that reported in Hayman et al. (2024a) captured at 8 kHz and 5 ns resolution and processed to 50 Hz and 75 cm resolution) we do not believe that this represents a *fully* comprehensive description of why most clouds produce non-physical water vapor retrievals. Nevertheless, masking data that produce errors is a necessary

component of DIAL processing. The masking approach described here represents a heuristic approach based on the described reasoning. Although masking would likely be further improved if more comprehensive understanding of these errors were robustly demonstrated, the approach outlined here represents an improvement over the methods we had described previously.

---

[2]the presentation is available at minute 32 here https://ams.confex.com/ams/103ANNUAL/meetingapp.cgi/Session/63551

These cloud biases tend to be more problematic with PTV retrievals than standard algebraic inversions, because, left unmitigated, clouds generally dominate regularizer selection. As such, they tend to require very low values of water vapor regularization (to support the potentially large swings in apparent water vapor in clouds). Those low regularizer values provide poor noise suppression in non-cloudy regions where data are valid. Ideally, we would leverage a noise model ($\mathcal{L}$) that inherently accounts for, and optimally trades, the potential errors imparted on lidar signals in clouds; however, none have yet been developed.

In order to enable PTV retrievals with the current noise model, we require a means to identify and mask potentially problematic regions prior to processing. We have previously outlined methodologies based on fluctuations in the signals (Hayman et al., 2024b). In this work, we have adopted a slightly modified method using a bootstrapping approach to estimate the inherent temporal variability of the observed fluxes and how they affect the variations in the noise model.

The masking approach we implement here attempts to capture the problem with the currently employed noise model by measuring a variation in the computed negative log-likelihood between independent samples. For each channel, the observed 2 second accumulated profiles are randomly split into two sets of photon counts. Each set of photon counts is then evaluated against the first set. This provides an estimate of the difference in the NLL to the best possible value (the first set evaluated against itself) and the independently sampled instance. In cases where the noise model properly accounts for the statistical variations (the temporal variations are not large and so a product of Poisson point processes), the differences will be relatively small. However, cases where the noise model incorrectly attributes high certainty will tend to produce large differences.

In implementation, we also treat the backgrounds between the two datasets as independent but known, and as such, observations in the independent set (designated set 2) have their background subtracted and the background from the evaluation data (designated set 1) added to them. The NLL calculation is then repeated several times with different random sampling to obtain a pseudo-estimate the variance in the NLL.

$$\sigma_{\mathcal{L}}^2 \approx \frac{1}{N-1} \sum_{n=1}^{N} \left[ \mathcal{L}(y_1^{(n)}, y_1^{(n)}) - \mathcal{L}(y_1^{(n)}, y_2^{(n)} - b_2^{(n)} + b_1^{(n)}) \right]^2 \tag{4}$$

where $y_1$ are the photon counts from the evaluation set, $y_2$ are the photon counts from the independent set, $b_1$ is the background of the evaluation set and $b_2$ is the background of the independent set.

The calculated variance in (4) is then used as the input argument to a logistic function similar to that employed in Hayman et al. (2024b) to provide a continuous deweighting of image pixels with high variance in the NLL. This method focuses on the suspected root cause of errors in clouds. It is therefore not directly tied to the presence of clouds in a pixel, but the underlying signal conditions causing errors in the NLL calculation which are common in cloudy pixels. As a result, other possible errors source (e.g. highly dynamic aerosol structure) can also be deweighted or masked if they are likely to contribute errors in the retrieval.

Note that this variance calculation accounts for effects of temporal variability to some extent (preliminary work suggests that MPD's observational cadence of 2 seconds is still too long to fully encapsulate variability in clouds), but does not account for accumulation over range variations. For this reason, there could be further improvement over the method employed here. However, the method has been more robust than our previously employed methods and has generally increased data availability.

In addition to masking due to potential errors in the noise model, the long pulse channels tend to experience a bias in the lower altitudes associated with the pulse length and recovery time of the detector. The way these potential errors impact biases in the instrument and ultimately impacts the minimum altitude of data products is discussed in detail in Spuler et al. (2021). These errors are in part due to the fact that data is not valid while the pulse is exiting and thus the long $1\mu s$ pulse blanks the receiver (which is electronically gated during the pulse exit) for 150 m. However errors persist in the recovered signals beyond this time and do not appear to linearly scale with the laser pulse length. The exact causes of this "recovery time" effect are still not fully understood and may be related to stray light, detector recovery time after exposure to high optical intensity (while gated), afterpulsing, transient effects of detector gating or a combination of all four. Ultimately, sloping baseline biases and transient detector responses can cause errors in the retrieval, particularly at low altitudes where backscatter signals from the atmosphere are relatively weak due to low overlap (see Figure 11 in Spuler et al. (2021)). The recovery time after the pulse exits does not appear to extend as high with the shorter pulse length. This effect is not encapsulated in the NLL so a heuristic solution of masking data below 500 m on the long-pulse observations and below 65 m or 80 m (for 100 ns and 200 ns pulses, respectively) on the short-pulse channel is employed.

### 2.3 Uncertainty

Some measure of uncertainty is needed to quality control remote sensing data. The best method we have found for lidar analysis uses bootstrapping and random manual thinning of the observational dataset. In this approach, sources of uncertainty are randomly varied and the variance of the data products are repeatedly calculated as described for our direct inversion DIAL processing in (Spuler et al., 2021). However, this requirement to repeat the processing increases computational expense. As a quick assessment of uncertainty, we take a simplified approach in which the processing is repeated only once but using different initial conditions. The concept is that if the optimizer does not converge on a value for a particular pixel, it likely indicates that there is higher uncertainty in the product. For the DIAL retrievals presented here, each scene is processed using two initial conditions: low value (0 $g/m^3$) and high value (10 $g/m^3$). These are selected because they provide a considerable spread in initial conditions, so it is relatively easy to identify regions where the solutions do not converge. This large difference in initial conditions also helps ensure that the data reported is well supported by the measurements and does not represent a non-unique solution dependent on the selection of initial conditions.

### 3 Experiment

The processing method described above is applied to the data collected from the MicroPulse DIAL (MPD) developed and operated by NSF-NCAR (NCAR/EOL-MPD-Team, 2020; Spuler et al., 2015, 2021). NSF-NCAR currently has five of these instruments which leverage diode laser technology for the transmitter source, and thus can implement various pulse configurations that change on a shot-to-shot basis. The hardware developed to enable multiple pulse configurations is described in Stillwell et al. (2025) and has been tested on one of the units. In most cases, long pulses are 1 $\mu s$ and short pulses are 200 $ns$. During one day in the data presented here (Nov. 1 2024), the short pulses were set to 100 $ns$. The long and short pulses

are fired in alternating sequences where the total pulse repetition frequency is 8 kHz and each individual channel (online-long, offline-long, online-short, offline-short) is thus repeated at 2 kHz. Photon count histograms are acquired at a base resolution of 2 s in time and 50 ns in time-of-flight (i.e., range). That data is further integrated in time during processing to a resolution of 10 s for backscatter-only estimates and 5 minutes for water vapor DIAL retrievals.

## 3.1 Backscatter

This first demonstration of processing multi-pulse-length data to retrieve the backscatter flux $\tilde{\rho}(r)$ is illustrative of how the retrieval can successfully leverage smeared observations with higher SNR using long pulses (1 $\mu s$) in combination with sharper lower SNR observations using short pulses (200 $ns$) to recover fine-scale atmospheric structure with low noise contamination.

The observations consists of a two hour time period where photon counts from MPD's WV offline channels are gridded to a resolution of 10 seconds and 7.5 m resolution. The scene consists of cirrus clouds extending up to 12 km while mid-level liquid clouds around 7 km are occasionally present with very fine-scale features. The estimated photon flux $\tilde{\rho}(r)$ resulting from processing this scene using the algorithm described here is shown in the top of Fig. 1 while the photon fluxes directly estimated from the observed photon counts in the offline-long and offline-short pulse channels are shown in the middle and bottom images.

The long pulse observations show higher photon count rates which result in higher SNR and allow better capture of high-altitude cloud features, but fine-scale structures like the liquid clouds are blurred significantly. The short pulse observations do much better at resolving fine-scale features in the scene, but suffer from low SNR, particularly in the high-altitude cirrus. The PTV retrieval using both of these observations is able to leverage the benefits of each. Fine-scale features like the water clouds are well resolved, while the high-altitude cirrus structure is also captured. In addition, the overall denoising capability enabled by PTV improves image quality by significantly reducing shot noise contamination, over both the long- and short-pulse direct observations.

By implementing the proposed PTV processing with forward modeling, there is no need to heuristically blend the two separate observations. Instead, only one estimate of the photon flux is obtained, which optimally describes the combination of long- and short-pulse measurements.

Note that the varying levels of noise in the direct estimates are the result of thinning 2 second profiles accumulated over a span of 10 seconds where the number of profiles used to accumulate a given 10 second bin vary. Because manual thinning is performed to obtain a validation dataset (for regularizer optimization), the number of 2 second profiles profiles in a ten second bin will vary between 2 and 3. This causes a striping appearance in the noise of the images, but does not affect the PTV estimate.

An example of the forward model projection onto the observed photon counts from this scene is shown in Figure 2. The smearing due to the different pulse lengths is encapsulated in the forward model. As a result, the thin liquid cloud layer at 7 km appears as a rect function in the long-pulse data and a narrow spike in the short-pulse data. The higher SNR from the long-pulse data helps recover information in sparser photon count regions. Photon flux is still recovered in regions where photon counts

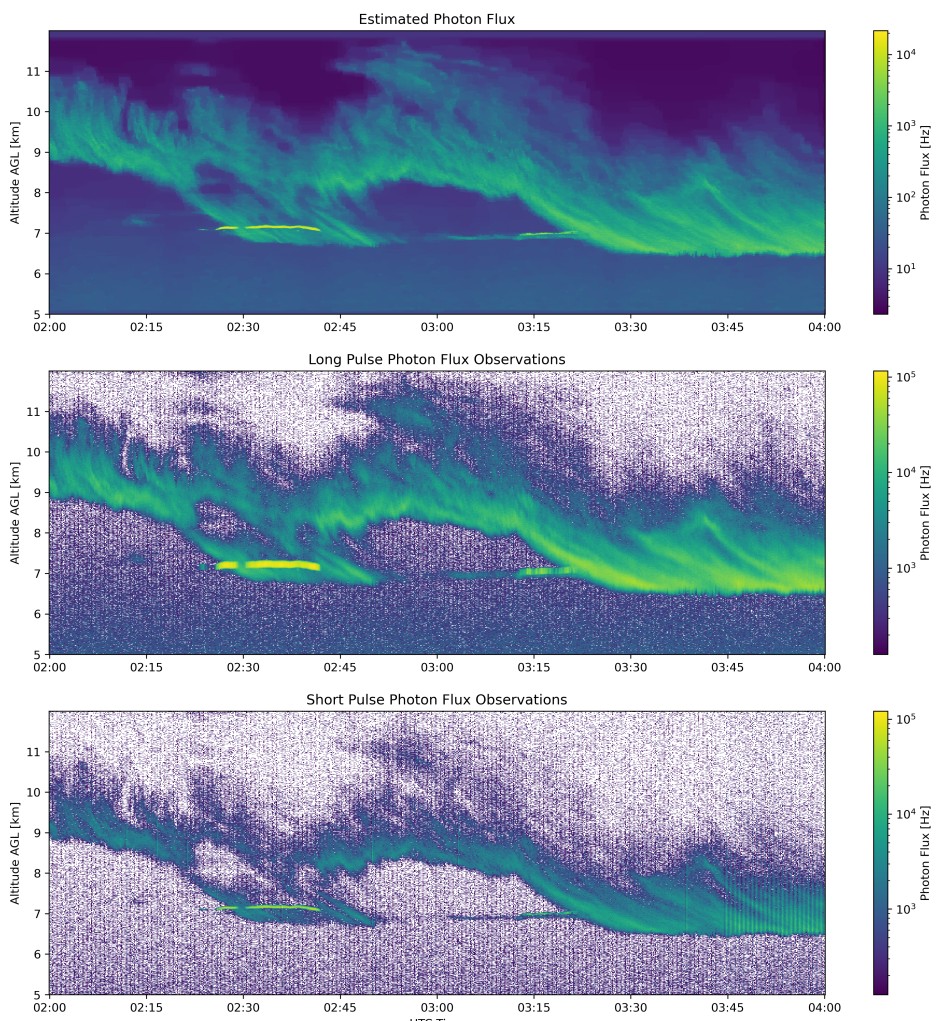

**Figure 1.** Cloud observations from May 7, 2024. The top panel shows the estimated photon flux using PTV with long and short pulse, offline channels. The middle figure shows the directly estimated flux from long pulse offline photon counts and the bottom shows directly estimated flux from offline short pulse photon counts.

are in single digits. As was shown in Hayman et al. (2024a), information content can still be obtained in regions with low photon counts because PTV leverages information based on the photon count clustering in addition to the counts in each bin.

### 3.2 Water Vapor

While backscatter observations are illustrative of how the proposed PTV algorithm can leverage multiple complimentary observations of the same scene, the actual scientific value of such observations is relatively limited. In fact, the principal motivation for developing multi-pulse-length configurations for MPD is to enable lower-altitude retrievals. The central issue with obtain-

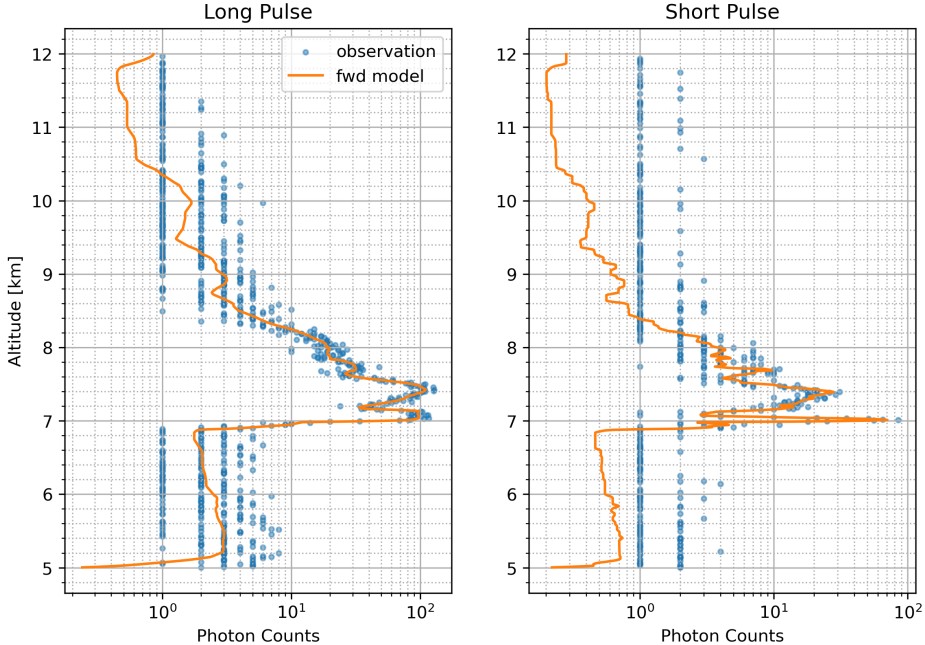

**Figure 2.** Estimator fit to photon counts observed in Long and Short pulse data at 3:20 UTC in Figure 1.

ing low-altitude retrievals from MPD is that the instrument is unable to resolve atmospheric returns for the duration of the pulse plus some recovery time (the recovery time is believed to be a combination of time for the detector to reenter a linear regime and decay of scattered light in the instrument). This is due to a design trade, where the instrument's coaxial transmitter, which enables high mechanical stability to accommodate a narrow field-of-view, which critically minimizes solar background contamination, also results in the receiver being blind during the outgoing pulse duration. By transmitting a shorter pulse, the instrument can observe at lower minimum altitudes.

For this evaluation, the raw data are binned at 5 minute time resolution and 7.5 m (50 ns) range resolution with observations between 50 m and 6 km. In standard (long pulse) operation, the minimum altitude for the MPD is between 300-500 m depending on the alignment and amount of recovery time needed by the detector. However, short pulses generally enable retrievals starting at 100 m, due to the shorter duration and a shorter recovery time as a resulting from the reduced transmitted energy. As such, the data obtained from long pulse mode are masked below 500 m, while short pulse data are masked below 65 m or 80 m (for 100 ns and 200 ns pulses respectively). In the backscatter example, we relied on the detector noise model to handle the trade-off between observational channels and fitting priority. However, since the noise model we use does not account for the errors contaminating low-altitude returns (the sources of which remain somewhat speculative), we resort to this manually masked approach.

During the fall of 2024, one of the MPD instruments (unit 4) was operated at the Marshall field site south of Boulder, CO, USA. During the operation period, nine radiosondes were launched on seven different days. This data was processed

using the PTV method described above. The processed data consisted of all four channels (online/offline wavelengths with combined long/short pulses), only short pulses, and only long pulses to assess if multi-pulse-length data provides a benefit over a single pulse configuration. We hypothesize that multi-pulse-length processing should provide retrieval performance that is comparable to the best of the two individual pulse configurations at any given altitude or time.

Because of its potential to significantly reduce computational expense, we also include combined processing with fixed regularizers in this analysis (denoted "Combined Fixed Reg."). These fixed parameters were defined on the basis of typical values obtained from other processing instances. The regularizers are not optimized on a scene-by-scene basis, which significantly reduces the number of processing instances that need to be run.

An example absolute humidity scene consisting of two days (October 28 - 29, 2024) is shown in Figure 3 with the combined retrieval, short pulse retrieval, long pulse retrieval, combined with no regularizer search and ERA5 Reanalysis (Copernicus Climate Change Service (C3S)) from top to bottom.

As per the masking scheme, the retrievals using short-pulse data extend lower, near 100 m above the the surface, where using long-pulse data in this region is not possible because the pulse is 150 m long and still exiting the instrument. That ability to observe closer to the surface enables the MPD to capture a dry layer below 500 m on Oct. 29 that is otherwise difficult to observe with only the long-pulse data. In higher altitude regions, the combined retrieval has similar data availability, ability to resolve structure (PTV will compensate for low SNR by integrating over larger patches) and noise suppression to the long-pulse data. Thus, in this particular example, it would seem that the combined retrieval blends the benefits of both pulse lengths.

The figures also highlight the substantial difference in water vapor structure captured by the ERA5 reanalysis and what is observed by the MPD. While structure is largely smeared out (in part due to the fact that the reanalysis is lower time and altitude resolution than the MPD), we also note that in this example ERA5 tends to overestimate absolute humidity near the surface and does not resolve the dry layer just above the surface on Oct. 29.

An example of the forward model fit to the raw photon count data from Figure 3 is shown in Figure 4. The color of each data point shown indicates the weight ($w_{c,\lambda,i}$ in Eq. (3)) which is used to de-emphasize or completely mask regions where the loss function is not believed to be valid. For example, below 500 m, all long pulse data is masked and therefore the data points are purple. Note that below 500 m the long pulse channels have poor agreement with the data. This indicates that the forward model or noise model are not consistent representations of the instrument operation between the two pulse lengths in this altitude region (highlighting the benefits of over constrained retrievals). By contrast, other altitude regions fit both channel well, indicating that our physical understanding of the captured signals (and therefore forward and noise models) is consistently accurate across both pulse lengths. The forward model is described by Eq. (1) and it's projection onto photon counts while accounting for deadtime by multiplying the estimated flux by active time $z_{c,\lambda,i}$.

## 4 Results

Nine radiosondes were launched while the MPD operated in multi-pulse-length mode at the Marshall Field Site, Boulder, CO, USA. These sondes provide some level of truth for evaluating the MPD retrievals. Figure 5 shows each radiosonde profile

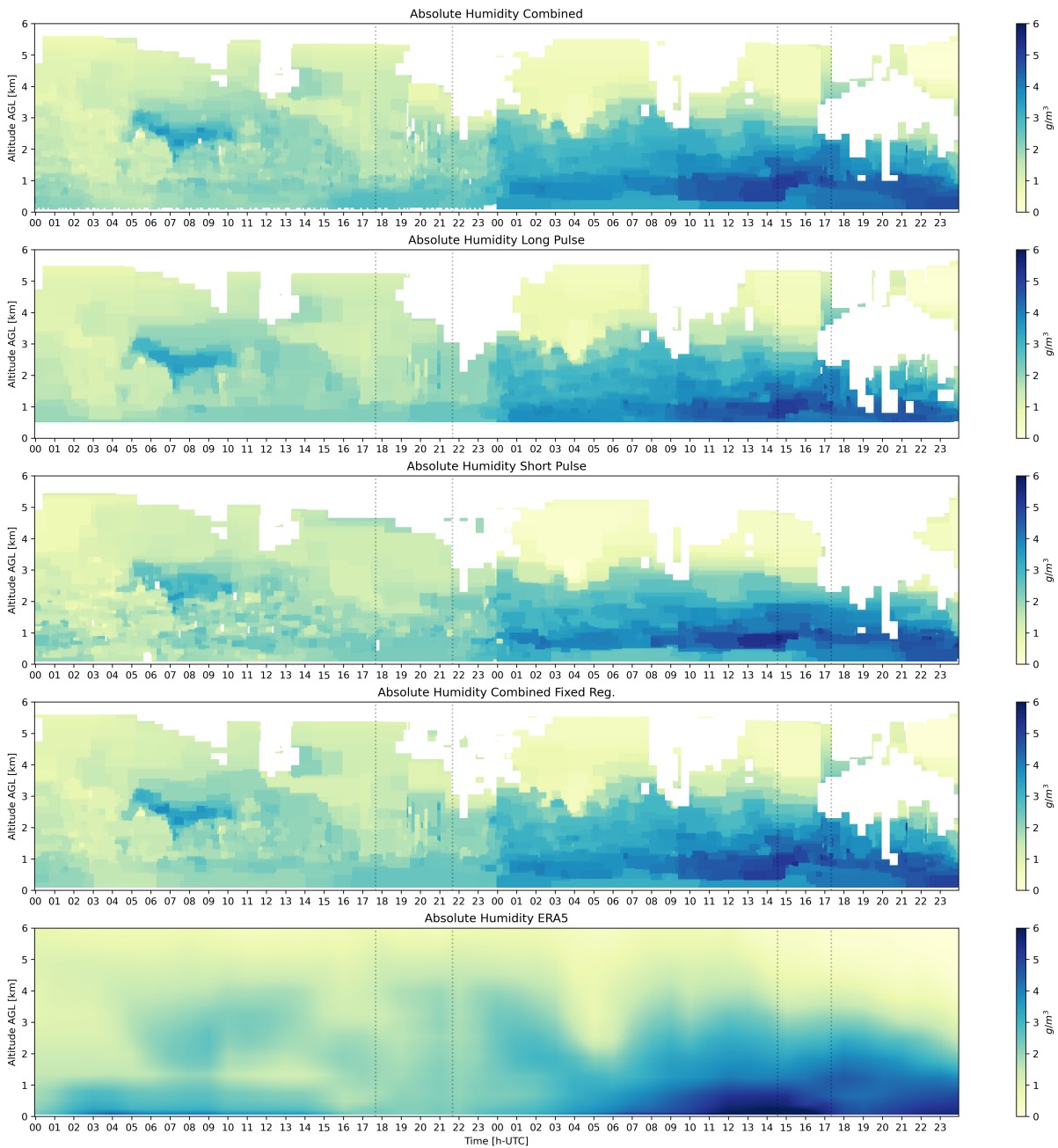

**Figure 3.** Absolute humidity estimates for October 28 and 29, 2024 near Boulder, CO, USA. The absolute humidity is shown for (from top to bottom) processing of both pulses, only the long pulses, only the short pulses, both pulses without optimizing the regularizers and ERA5 reanalysis interpolated onto the MPD location. White regions are masked due to pulse length, clouds or poor convergence in optimization. Dashed lines indicate times where radiosondes were launched.

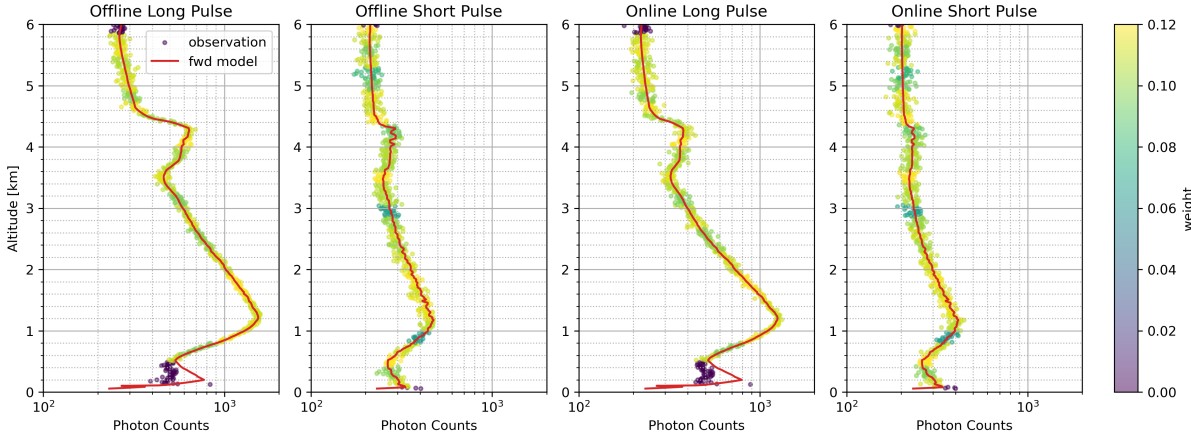

**Figure 4.** Forward model projection onto photon count data of all four channels at 14:00 UTC on October 28, 2024. The weight of each data point is shown by it's color with masked points appearing purple.

with the MPD absolute humidity estimates, where the approximate uncertainty is indicated by the shaded region. Retrievals are shown for each of the four processing instances. In addition, we show the ERA5 reanalysis (Copernicus Climate Change Service (C3S)) estimate of absolute humidity at the MPD location to help assess if and where the instrument may provide

information about atmospheric state that is not captured by the existing reanalysis product. If the model accurately represents all observed atmospheric structures, this would imply that the observations do not add information content, and by extension, value.

Overall the MPD retrieved profiles tend to capture the water vapor structure fairly well. The short pulse data tends to deviate from the sonde more at higher altitudes and generally exhibits higher uncertainty than data making use of long pulse data. The

long pulse data is not available below 500 m due to biases in the signal below these altitudes. Notably, by leveraging the short pulse data, the MPD retrieval is able to capture a dry inversion layer below 500 m on 2024-10-28 14:33:07 as can also be seen in the example data in Figure 3. This inversion is otherwise missed when using long pulse data and the ERA5 data entirely misses the feature. On the last sonde (2024-11-01 16:31:30) the short pulse length was 100 ns. In this dataset, the short pulse and the combined retrieval (with optimized regularizer) are both very noisy. This suggests that some of the processing criteria

are not necessarily well tuned for this configuration (e.g. masking criteria and minimum altitude of long pulse). The forward model of the 100 ns short pulse may also be less accurate as the rise time of the amplifier output is not necessarily negligible at this short of a pulse length. However the excessive noise likely stems from improper regularizer selection, as the combined processing using a fixed regularizer is not nearly as noisy and appears to recover the water vapor structure relatively well. This demonstrates how optimization of the regularizer may not always produce the best results when corrupted data is used to

estimate the optimal values. It also demonstrates how further work on establishing optimal pulse length combinations would be highly useful for establishing the best mode of operation for this multi-pulse-length technique.

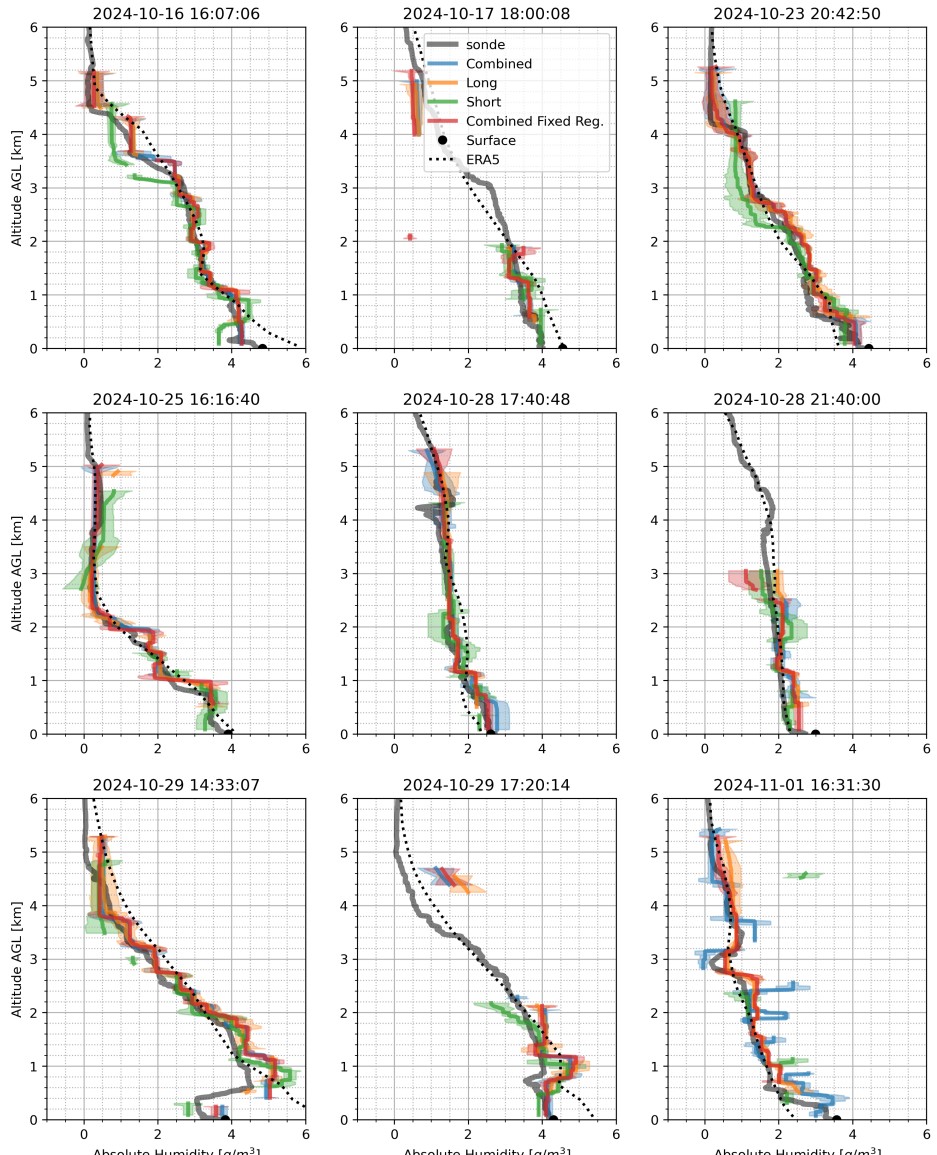

**Figure 5.** Comparison of absolute humidity estimates from MPD using combined (blue), long (orange), short (green) and combined without regularizer optimization (red), and with radiosondes (gray) launched during the Marshall test in October/November 2024. Also shown are the ERA5 estimates of absolute humidity (dotted black) interpolated to the MPD location.

Performance statistics of the absolute humidity estimates are conducted by interpolating the estimate onto the sonde's higher resolution altitude data (ensuring that lower resolution retrievals are appropriately penalized). The root-mean-square-difference, mean difference, standard deviation of difference and correlation coefficient are then calculated for all estimates using the sonde absolute humidity as reference. We also calculate and report the data availability. For the purposes of this aggregate analysis,

MPD data with uncertainty greater than 0.5 $g/m^3$ are masked in the data quality control process. It should be recognized that by making masking more aggressive, most error analysis statistics will improve (assuming effective error metrics). Thus, it is critical that we also include metrics on data availability to balance the masking criteria.

A summary of these metrics for all radiosonde launches is shown in Table 1. Over the full 0-6 km range, the best retrieval is obtained from the combined processing with regularizer optimization. However, we note that the combined processing without regularizer optimization performs nearly as well and has the highest data availability. In the lowest 1 km, the combined fixed regularizer also has the highest data availability and the lowest RMSD but is slightly out performed by the long pulse in correlation coefficient. However this improvement in correlation in long pulse data is likely the product of simply not capturing lower altitude structure (it has the lowest data availability). In this aggregate analysis for both altitude ranges, ERA5 has lower RMSD and correlation coefficients than all MPD retrievals; however, it has full data availability. The lower performance of ERA5 is most pronounced in the lower 1 km analysis, highlighting a potential area of weakness for the product in an area that maybe very important for high impact weather analysis.

| Range | Product | data points | RMSD | mean | std. | Correlation |
|---|---|---|---|---|---|---|
| 0-6 km | | | | | | |
| | Combined | 8499 | 0.33 | 0.16 | 0.28 | 0.980 |
| | Long | 7824 | 0.33 | 0.18 | 0.27 | 0.978 |
| | Short | 6420 | 0.37 | 0.06 | 0.37 | 0.962 |
| | Combined Fixed Reg. | 8931 | 0.34 | 0.14 | 0.31 | 0.975 |
| | ERA5 | 13478 | 0.42 | 0.17 | 0.38 | 0.959 |
| 0-1 km | | | | | | |
| | Combined | 1647 | 0.47 | 0.35 | 0.31 | 0.936 |
| | Long | 1307 | 0.45 | 0.36 | 0.25 | 0.967 |
| | Short | 1700 | 0.46 | 0.22 | 0.40 | 0.893 |
| | Combined Fixed Reg. | 1808 | 0.44 | 0.33 | 0.29 | 0.953 |
| | ERA5 | 2365 | 0.81 | 0.39 | 0.72 | 0.840 |

**Table 1.** Analysis of absolute humidity estimate differences with radiosondes over ranges from 0 to 6 km and 0 to 1 km

A range-resolved aggregate analysis of MPD products and ERA5 compared to all nine sondes is shown in Figure 6. Here we plot the fraction of data available as a function of altitude (note that ERA5 data availability is not shown as it is 1.0 in all cases) as well as range-resolved RMSD and correlation coefficients. Both combined retrievals tend to have data availability comparable to the long-pulse cases at higher altitudes, but comparable to short-pulse instances at lower altitudes. For most cases where data is available, the RMSD is comparable across all the pulse length configurations. Between altitudes of 500 m and 4 km, the absolute humidity of ERA5 has similar RMSD to the MPD processing with a slightly lower correlation coefficient. However below 500 m the ERA5 has noticeably larger RMSD and lower correlation coefficient. Above 4km, the MPD correlation coefficients drops below that of the ERA5 and continues to drop in altitude while the RMSD remains somewhat

similar to the ERA5. Note that at higher altitudes, absolute humidity tends to be lower, so RMSD will naturally tend to decrease if the signal processing does not become excessively noisy. This also means achieving high correlation coefficients at these altitudes requires very high sensitivity to changes in absolute humidity which may be below the uncertainty of the instrument at these higher altitudes. It should be recognized however, that the MPD is still well suited for capturing atypical (non-canonical) cases of higher absolute humidity at higher altitudes which may be missed or poorly resolved by models/reanalysis and passive profilers. An example of this is seen in Figure 3 between 5 and 10 UTC at 3 km. This feature is not included in the aggregate analysis as no sondes were launched during that time.

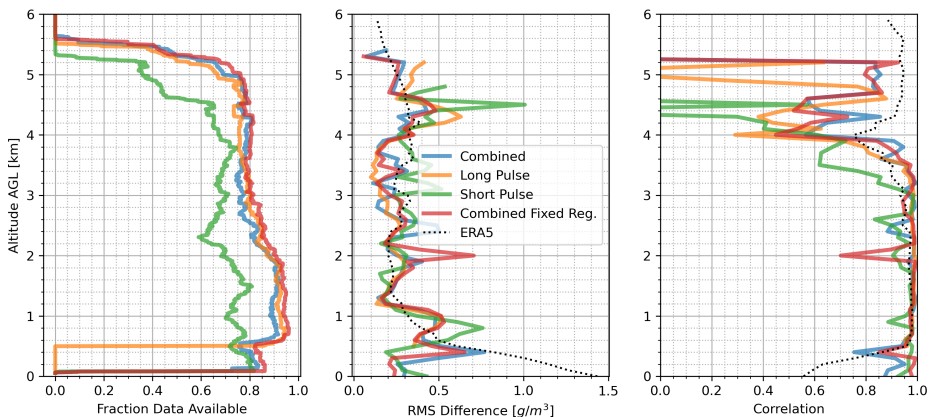

**Figure 6.** Marshall field test data availability during sonde launches (left), root mean square difference of absolute humidity estimates with sondes (middle) and correlation coefficient between estimate and sonde (right). MPD profiles shown for combined (blue), short (orange) long (green) and combined fixed regularizer processing (red) cases. ERA5 renalysis results are shown as a black dotted line.

The aggregate analysis supports the claim that a joint retrieval of absolute humidity with both long and short pulse data captures the best of both approaches. This combined approach enables low-altitude data collection (below 500 m) and comparable or better data availability throughout the 6 km profiles as either pulse configuration.

While this analysis has focused on treating 1 $\mu s$ pulses as "long", we should note that this pulse length was originally selected as a reasonable compromise between minimum altitude and SNR. As such, it might be the case that even longer pulses would be more advantageous and provide a better compliment to the short-pulse observations. Furthermore, there is also no reason to expect that a combination of only two pulse lengths would be optimal. The hardware developed for this capability and the signal processing presented here could be applied to a variety of pulse length combinations and are not restricted to just two. Finally, this work makes use of individual square laser pulses; however, similar processing could be applied to a variety of pulse shapes and sequences. In short, there is considerable need for further research into lidar pulse configurations to maximize the capture of information content from atmospheric scenes.

In baselining MPD results against ERA5, it should be recognized that the sonde sampling represents a small dataset and most of the cases considered here do not represent complex or high-moisture scenes that are difficult for models to resolve, nor

are they necessarily associated with high impact weather. Assuming that the MPD profiles of absolute humidity structure are representative of the true atmosphere, Figure 3 serves as a demonstration of how the ERA5 may miss significant amounts of structure in the water vapor field and have substantial errors particularly at the lowest altitudes. Recent work showed that low vertical resolution observations (equal or coarser than 500 m) "are not sufficient for profiling with the intention of calculating PBL convection parameters" (Hoffman and Demoz, 2025). The question as to whether ERA5 is a sufficiently accurate representation of the water vapor field likely depends on how the data drives a particular analysis and how smaller-scale features, or vertical distribution of water vapor, impact the results. ERA5 reanalysis is frequently used in case studies attempting to identify the atmospheric state that leads to convection (e.g., Tuckman and Emanuel (2024)) and has been adopted as a standard for the field in training AI numerical weather models (e.g. Schreck et al. (2025); Price et al. (2025)). The results presented here (particularly the example shown in Figure 3) raise some questions about whether over reliance on convenient products (high coverage and data availability) could eventually hinder or impose a performance ceiling on developments in atmospheric science and Earth system prediction. The capability to continuously profile at horizontal and vertical resolutions capable of capturing shortcomings in models and reanalysis (where and why they fail or succeed) is likely a key step in advancing models of the Earth system beyond current limitations.

## 4.1 Optimization time

Processing a single day with a full regularizer search took between 4-10 hours depending on resource availability (number of GPUs) on the Casper computing cluster, termination criteria, and the complexity of the observed scene. The computational cost to fully process a day of data ranges from 50-200 GPU hours. However, if we omit the regularizer optimization step, this significantly reduces processing time and cost. Typically, processes on the GPU are completed in less than two hours. For the scenes considered here, the median GPU hours used to process the combined (using both long- and short-pulse data) data is 145. However, for fixed regularizer (which has been selected based on typical values), the median GPU hours are 0.86. Note that all processing hours calculated include two processing runs for uncertainty estimation, as described before. A plot showing the total GPU processing hours for all cases considered here is shown in Figure 7. While processing cost can be significant, omitting the regularizer search significantly reduces this cost.

The PTV processing routine can run on CPUs but with significantly reduced speed compared to GPUs (highly parallel GPU architecture is well optimized for accelerating image processing routines). The speed of processing varies significantly depending on the number of CPUs employed. We found with some cursory benchmarking that a single CPU process is relatively slow with the PTV optimization running at 0.03 iterations/second. Four CPUs increase the speed to 0.2 iterations/second and eight CPUs 0.3 iterations/second. This is in contrast to the GPU execution speed which is 3-6 iterations/second. Thus, there is clearly a significant speedup achieved with GPU hardware, but executing daily processing without optimizing regularizer can be performed on CPU-based servers. Processing a single day's worth of data with fixed regularizer settings would typically take about 8 hours (assuming no parallelization for multiple runs to calculate uncertainty), which is fast enough to keep up with the daily data stream.

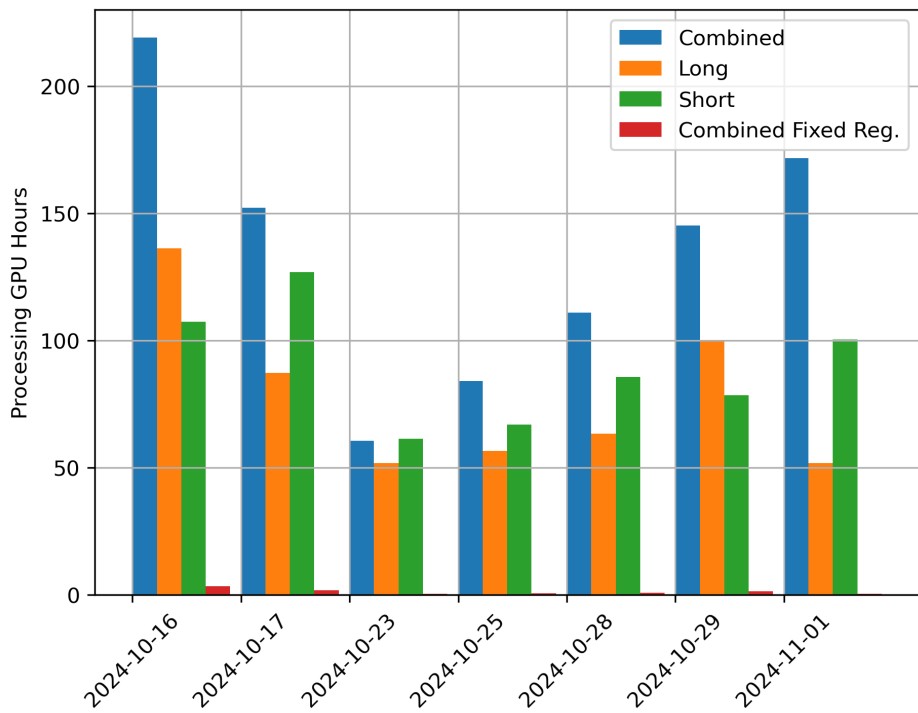

**Figure 7.** Processing cost in GPU hours for all dates considered in this work using different approaches. Processing considered here are based on what channels are processed (Combined, long pulse data, short pulse data) and whether a regularizer search is performed.

All processing that includes regularizer optimization was conducted on the Casper computing system operated by NSF-NCAR which, at the time of this work, has 52 Nvidia V100 GPUs. The number of GPUs available to parallelize the regularizer search depends on the amount of GPU use from other users and can vary throughout the process.

Processing without regularizer optimization is conducted on a standalone server which has two Nvidia Quadro RTX 5000 GPUs. Having two GPUs enables simultaneous processing of both initial conditions for uncertainty estimation. There does not appear to be a substantial difference in processing speed or performance between the Quadro and V100 hardware models.

## 5 Conclusions

In this work we demonstrate an approach to denoising and inverting absolute humidity from an MPD transmitting alternating long ($1\mu s$) and short ($200ns$ or $100ns$) laser pulses. In doing this, we are able to obtain water vapor estimates at lower altitudes than is possible with the longer pulses, but also at higher altitudes than are possible with just the short pulses. Through a validation analysis with sondes, we show that the combined method has comparable performance to the better of the two individual pulses. We also note that the selection of pulse lengths for this analysis was fairly arbitrary. Further investigation

of the optimal pulse lengths (including the number of different variants) may further improve the performance of the retrieved products.

A baseline comparison to ERA5 reanalysis shows the current combined MPD retrieval produces a clear water vapor profiling improvement below 500 m with comparable aggregate performance above this altitude. This highlights the importance of the new short pulse observations as this region is difficult to interrogate with the long pulses alone. However the example data also highlights how reanalysis may struggle with atypical water vapor structure and vertical distribution. For the limited cases shown here, the benefits of MPD water vapor observations are most apparent in the lowest altitudes. However, models will likely need other corrections from observations, and the specific factors influencing this need are not presently well known. Given that ERA5 has been adopted as a standard for the field as training and validation data in AI numerical weather models, it raises some questions the how that practice may impose a limit the performance of those models.

PTV processing has consistently been shown to produce more accurate data products than standard inversions. In this case, PTV also merges products that would otherwise be very difficult to combine due to the variable pulse length structure. However, PTV performance typically comes at the cost of longer processing times. In this work, we show that some performance sacrifices can be made to significantly speed up the PTV processing and make it practical for rapid delivery of results. By omitting optimization the regularizer to the scene, processing of an entire day typically completes in less than an hour. The omission of this optimization step is equivalent to setting a fixed smoothing kernel across scenes, which, to our knowledge is more common than optimizing them on a scene-by-scene basis. Our analysis suggests that this fixed regularizer approach to processing MPD data performs nearly as well as the full processing approach (in some cases better such as on Nov. 1, 2024) when the regularizer parameters are chosen well.

Overall, this work demonstrates the advantages of diode-laser flexibility in lidar architectures. In spite of their relatively low peak powers, it is possible to operate diode-lasers and process the data to produce high quality, low error estimates of water vapor from 100 m up to 6 km. This, in combination of a number of practical benefits such as cost, eye safe transmitters (class 1M) and invisible wavelengths makes such approaches well suited to potential large scale network deployment. By developing complimentary hardware configurations and signal processing approaches, we have shown clear benefits in both water vapor retrievals and backscatter estimates. Development of these techniques may enable high performance observational capabilities on airborne and space borne platforms with instruments of lower size, weight and power.

*Code and data availability.* MPD data is cataloged in the NSF NCAR Earth Observing Laboratory's Field Data Archive with the following URL and digital object identifier: https://data.eol.ucar.edu/dataset/100.034 and https://doi.org/10.26023/MX0D-Z722-M406. (NCAR/EOL-MPD-Team, 2020).

The optimizer used for regularized maximum likelihood estimation in this work has been made publicly available at https://github.com/NCAR/SpiralTorch

*Author contributions.* M. H. developed and ran the signal processing algorithm and performed data analysis. R. S. and A. K. developed and integrated the electronic hardware to shot-to-shot modify MPD laser pulse length. A. K. wrote the custom CUDA kernel implementation of FISTA. S. M. developed the original MPD concept, aligned the instrument, investigated optical sources of error in the instrument products. All authors contributed to writing and editing the manuscript.

*Competing interests.* The authors declare that they have no competing of interests.

*Acknowledgements.* This material is based upon work supported by the NSF National Center for Atmospheric Research, which is a major facility sponsored by the National Science Foundation under Cooperative Agreement No. 1852977. We would like to acknowledge computing support from the Casper system (https://ncar.pub/casper) provided by the NSF National Center for Atmospheric Research (NCAR), sponsored by the National Science Foundation. AI (Gemini 2.5 Flash) was used to assist in the drafting of the article summary which was manually edited for tone and accuracy.

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
