# Peer review of "Signal processing to denoise and retrieve water vapor from multi-pulse-length lidar data"

_EGUsphere, 2025_

## Author Comment (AC1)

We want to thank the reviewer for their time and careful consideration of the manuscript. They have brought several important points to our attention. Reviewer comments are included below in **black** and our responses are printed inline in blue. *Italic* text is used to quote text in the manuscript.

**Reviewer 1**

I find that the author make sufficiently clear that their method of signal processing improves the results from multi-pulse-length LIDAR measurements, in that it enables to merge data obtained from longer and shorter laser pulses and extract a better picture of water vapor distribution in the atmosphere. Therefore I recommend this paper for publication. I have only minor issues that I hope the authors might clarify.

1) Abstract: It is not clear to me why longer pulses should blank the detector. If the energy is distributed over a longer time, the dynamic range of the detector should not be the limit. Please explain it better. Is this only due to the fact that no data can be recorded while the pulse is still on its way out of the laser? Or there are other processes involved (see following point)?

From a simplest, first order perspective the reviewer's description provides an intuitive model for understanding why shorter pulses are needed to see lower. The instrument is effectively blind as the laser pulse exits and therefore a longer pulse forces a higher minimum altitude. Please note that the laser pulses employed by the MPD are much longer than are typical with solid state lasers. Typical MPD laser pulses of 1 µs result in a blanking range of 150 m. However there appear to be some larger complexities that result in a longer time before accurate data can be recovered. At this point, we don't feel that we fully understand all the contributing factors. While we are willing to discuss these factors (and we continue to investigate them), we also want to be clear that they are also beyond the scope of this particular work.

To help clarify this, we have revised the final paragraph in the masking section to read In addition to masking due to potential errors in the noise model, the long pulse channels tend to experience a bias in the lower altitudes associated with the pulse length and recovery time of the detector. The way these potential errors impact biases in the instrument and ultimately impacts the minimum altitude of data products is discussed in detail in \cite{Spuler2021}. These errors are in part due to the fact that data is not valid while the pulse is exiting and thus the long \$1 \mu s\$ pulse blanks the receiver (which is electronically gated during the pulse exit) for 150 m. However errors persist in the recovered signals beyond this time and do not appear to linearly scale with the laser pulse length. The exact causes of this "recovery time" effect are still not fully understood and may be related to stray light, detector recovery time after exposure to high optical intensity (while gated), afterpulsing, transient effects of detector gating or a combination of all four. Ultimately, sloping baseline biases and transient detector responses can cause errors in the retrieval, particularly at low altitudes where backscatter signals from the atmosphere are relatively weak due to low overlap (see Figure 11 in \cite{Spuler2021}). The recovery time after the pulse exits does not appear to extend as high with the shorter pulse length. This effect is not encapsulated in the NLL so a heuristic solution of masking data below

500 m on the long-pulse observations and below 65 m or 80 m (for 100 ns and 200 ns pulses, respectively) on the short-pulse channel is employed.

We currently believe that potential contributing factors related to corruption in the backscattered signals at low altitudes are related to combinations of scattering in the instrument, detector afterpulsing, nonlinear optical amplifier response to drive current, residual deadtime errors, detector response to the gate and detector illumination while gated.

Factors such as scattering can cause the exit pulse to extend beyond the actual "on" duration due to increased optical path length, but we presently believe that scattering probably just dictates the *amount* of corrupting pulse signal coupled into the receiver. Extended tails in the response are believed to be the product of afterpulsing in the detector AND nonlinear responses in the optical amplifier (after the current is turned off, similar to afterpulsing, trapped carriers recombine to continue to emit a weak, decaying signal which can be scattered into the receiver). These tails, because they are not constant in range, and atmospheric signals are comparatively low, can cause biases in DIAL retrievals. Gating the detector significantly decreases the amount of afterpulsing from the detectors, but the detector still may experience some recovery time after the gate is reenabled, resulting in a sloping baseline response effect. Additionally, the baseline response of the detector does appear to be influenced by the light exposure even while the gate is applied. Finally, deadtime, while corrected using more advanced methods (see Kirchhoff et al 2025) may not be adequately addressed due to insufficient range resolution at the steepest parts of the baseline response curves.

Many of these terms will tend to dominate the backscatter signals nearest the instrument where signal photon counts are low and thus contribute significant errors. Note that removing these terms relies on accurate estimates of the baselines, doesn't account for some of the stated potential factors, and poor correction can increase errors. The figure below shows some example data of a system transmitting 4 different laser pulse lengths along with the associated measured baseline calibrations (dashed lines). While it's clear that the shorter pulses produce valid data at lower altitudes, the exact location where that data is valid is likely beyond the point where the detector gate is turned on (something seen circumstantially based on the fact that the retrieval is biased).

In order to push retrievals lower, we will need to better understand these kinds of effects and how they can be characterized and modeled to provide corrections or uncertainty estimates. This is an area we are actively working on, but for now, leveraging a combination of multiple pulse lengths appears to help address the problem without comprehensive knowledge of it.

2) Page 7: "In addition to masking due to potential errors in the noise model, the long pulse channels tend to experience a bias in the lower altitudes associated with the pulse length and recovery time of the detector (the exact causes of this effect are still not fully understood and may be related to stray light, detector recovery time, afterpulsing or a combination of all three)."

How is the dependency of this effect on the pulse length. Can you give a quantitative answer?

As per our response above, we don't fully understand the factors involved. For example, it appears that the nonlinear response of the optical amplifier is different for different pulse lengths. But we have also noticed that the detector response after the gate is reenabled seems to change based on the amount of illumination while the gate is disabled. To avoid over speculation, we have not made any changes to the manuscript in response to this comment.

3) Figure 5, bottom right panel. The authors discuss the noisy data (blue line), but the average value of the blue line seems to be much shifted toward higher humidity as well. Can they explain it?

As discussed starting at line 313 in the text, the retrieval appears to suffer from poor regularizer selection on this day, possibly due to improperly tuned masking criteria (highlighting a drawback of optimizing regularizers for every scene). We don't really agree that the blue line is "much shifted toward higher humidity". The visible data is approximately 0.5 g/m^3 higher than the other profiles, but a lot of the lower swings have been masked based on the QC masking criteria, so the masked view tends to be slightly biased.

We have added some clarifying language to better explain this near line 313. In this dataset, the short pulse and the combined retrieval (with optimized regularizer) are both very noisy. This suggests that some of the processing criteria are not necessarily well tuned for this configuration (e.g. masking criteria and minimum altitude of long pulse). The forward model of the 100 ns short pulse may also be less accurate as the rise time of the amplifier output is not necessarily negligible at this short of a pulse length. However the excessive noise likely stems from improper regularizer selection, as the combined processing using a fixed regularizer is not nearly as noisy and appears to recover the water vapor structure relatively well. This demonstrates how optimization of the regularizer may not always produce the best results when corrupted data is used to estimate the optimal values. It also demonstrates how further work on establishing optimal pulse length combinations would be highly useful for establishing the best mode of operation for this multi-pulse-length technique.

4) What happens if instead of short and long pulses one used only short pulses with more or less energy? I understand that this might not be possible with the present setup, but what if?

For improved signal to noise ratio, higher pulse energies are generally better, but we don't really know how this might alter the aforementioned corrupting terms that extend beyond the laser pulse length.

We are not sure what the reviewer asks is physically possible, but engaging in the hypothetical: One advantage of short pulses is that the pulse length need not be deconvolved from the observations if they are very short (standard Nd:YAG pulses being on the order of a few ns). In a perfect world, modulating the power but leaving the pulse length constant would make more standard image fusion techniques (combining lower power noisier pulses with higher power less noisy pulses) more practical. As it stands, PTV is doing double duty by both denoising the image and deconvolving the pulse width from the observation.

---

## Author Comment (AC2)

We want to thank the reviewers for their time and careful consideration of the manuscript. They have brought several important points to our attention. Reviewer comments are included below in **black** and our responses are printed inline in blue. *Italic* text is used to quote text in the manuscript.

**Reviewer 2**

This paper discusses the signal processing methods for combining short and long pulse signals from ground based micropulse water vapor DIAL system to reap the benefits of both in different signal regimes with a goal of enabling water vapor retrievals closer to the surface which is of interest the broader atmospheric science community. The paper is well written, and methods are adequately discussed. I recommend the paper to be published after revisions that land between major to minor.

**Minor Comments:**

Title: Consider adding ground-based to the title to better capture the focus of the manuscript (...from multi-pulse-length ground-based lidar data)

While this demonstration uses a ground based lidar, this is not an essential element of the work. On airborne platforms, there is a desire for high resolution, low noise observational data, constrained footprint and power consumption and observations close to the aircraft to bridge between remote and in situ observations. Also, as demonstrated in section 3.1, the approach has applicability to other types of lidar (e.g. HSRL) and can be used to provide high resolution retrievals at potentially lower Size, Weight and Power. While aspects of the code base would need to be modified for an airborne or spaceborne system, the technique is still applicable. We have attempted to expand on the generalizability of this work in both the introduction and conclusion to be clear that it is not restricted to a particular platform.

**Line 2. Add 'in duration' after (low pulse energy)**

**Done.**

There are many references to longer pulses saturating the detector and causing 'ringing' or non-linearities several microseconds after the 1 us pulse shuts off that limit the utility of the longer pulses for quantitative DIAL retrievals near the surface. The amount of light on the detector from the laser flash is proportional to the pulse width. If the dominant effect limiting the utility of the 1 us pulse is the after pulsing, you should be able to show the effects of after pulsing between the long and short pulses by blocking the transceiver and directly plotting the afterpulsing between the two configurations. This would be a useful figure to convince the reader of the root cause that resulted in this work. The effect of 'smearing', absent any

non-linear detector effect, that limits the near field water vapor retrieval seems like a stretch and should only limit the retrieval down to ~150 m above the surface. Having a figure demonstrating this root cause would be a good addition to the narrative.

The purpose of this work is to develop a means of processing multi-pulse-length data collected using hardware and technique described and published in Stillwell et al 2025. Specifically describing the root cause of the issues motivating Stillwell et al extends beyond the scope of this work. When discussing this, we are intentional about the fact that we do not claim to understand the root cause (also please see our detailed response to Reviewer 1, item 1.). This does not mean we don't think the root cause is important. These issues (note we believe they are more than afterpulsing or ringing per our response to reviewer 1) remain an active area of investigation, and we strongly feel that the community should devote more resources to understanding non-ideal behavior of instrument system components. While baseline calibrations are performed on the instruments (see Hayman et al 2024), we do not feel it makes sense to show these because they don't necessarily encapsulate the various potential issues at play in the time immediately following the pulse exit and we do not want to over emphasize a particular explanation based on speculation.

We have revised the final paragraph in the masking section to read In addition to masking due to potential errors in the noise model, the long pulse channels tend to experience a bias in the lower altitudes associated with the pulse length and recovery time of the detector. This is in part due to the fact that data is not valid while the pulse is exiting and thus the long \$1 \mu s\$ pulse blanks the receiver (which is electronically gated during the pulse exit) for 150 m. However errors persist in the recovered signals beyond this time and do not appear to linearly scale with the laser pulse length. The exact causes of this "recovery time" effect are still not fully understood and may be related to stray light, detector recovery time after exposure to high optical intensity (while gated), afterpulsing, transient effects of detector gating or a combination of all four. Ultimately, sloping baseline biases and transient detector responses can cause errors in the retrieval, particularly a low altitudes where backscatter signals from the atmosphere are relatively weak. The recovery time after the pulse exits does not appear to extend as high with the shorter pulse length. This effect is not encapsulated in the NLL so a heuristic solution of masking data below 500 m on the long-pulse observations and below 65 m or 80 m (for 100 ns and 200 ns pulses, respectively) on the short-pulse channel is employed.

Lines 45-50. Please elaborate how a direct DIAL retrieval is over constrained and to what four observations you are referring. If this is a reference to PTV, please make clear that the constraint comes about from the use of the PTV method.

By observing the same atmosphere with two different pulse lengths, we obtain four channels of observations (offline and online for each pulse length) which means there are 4 observations with 2 unknowns. This means the water vapor inversion problem is over constrained which no

longer lends itself to direct algebraic inversions. We have attempted to update the text in the manuscript to be more clear about this:

In \cite{Stillwell2025} the hardware development of this concept is described and the potential of the approach is demonstrated using a conventional water vapor DIAL retrieval (direct inversion with the DIAL equation). As such, the inversion of water vapor and backscatter fields is over-constrained (two unknowns -- absolute humidity and attenuated backscatter cross section -- with four observations -- online and offline wavelengths at two different pulse lengths). In this configuration, applying the DIAL equation produces two separate estimates of absolute humidity (one for each pulse length)...

Lines 48-55 – The authors have made a clear case of the utility of PTV for overcome limitations of low flux lidar systems, however, it's unclear why PTV is needed to merge data from long and short pulses for the near field water vapor retrievals. Why not use a weighted average with altitude to simple merge the two retrievals near the surface? Would this not be computationally less expensive? Lines 69-72 indicate that a 100 m resolution is used for the near surface retrievals...is PTV needed if a single vertical resolution is employed for the near field retrieval?

Lines 69-92 the comment regarding the resolution of the instrument is based on an approximate estimate of the effective resolution of the instrument, not a set resolution. Our assertion is that, in the context of the resolution, this is effectively observing to the surface. We have revised the text to clarify this.

We roughly estimate the effective resolution of MPD water vapor retrievals are about 100 m (not to be confused with a capture resolution of 7.5 m), thus suggesting the MPD is effectively observing water vapor to the surface.

Independently processing the long and short pulse data and subsequently merging them through a weighting function probably would not be more computationally efficient unless the processing were conducted using the standard DIAL retrieval. Using the standard DIAL equation and heuristically merging the two retrievals is shown in Stillwell et al 2025. Inspection of Figure 5 in that paper, shows that there tends to be some banding in the merge region between 500 and 750 m where the short pulse data appears to be more noisy in time than the data integrating the long pulse data above 500 m. Artifacts through direct merging are a common issue. Using a joint retrieval with all observation channels helps alleviate this by forcing consistency across all observations and avoids sharp discontinuities. It also continues to leverage information content from the short pulse data above the merge altitude. Finally, merging across known fixed boundaries is straightforward to implement, but the method employed here also tends to result in varying levels of masking near cloud boundaries, where short pulse data sometimes remains valid even if long pulse is not. The joint retrieval automatically handles these gaps in data availability from some channels, where manual merging in these regions would be complicated. The text in the manuscript has been updated to address these comments.

In this configuration, applying the DIAL equation produces two separate estimates of absolute humidity (one for each pulse length); which must be subsequently merged to produce a unified water vapor estimate. This manual merging process has some key drawbacks recognized in \tite{Stillwell2025} in that the artifacts can appear in the merge region between the independently estimated products (see Figure 5 of \tite{Stillwell2025}), information content is discarded and therefore retrieval precision is not as high as it could be, and the potential benefits of short pulses enabling recovery of high resolution features (more apparent in cloud/aerosol lidar) are lost. In addition, leveraging joint retrievals with over-constrained systems tends to act as a useful quality control check on the recovered data products because the inability to fit all data sets indicates inaccuracies or invalid assumptions in the instrument model used for the inversion algorithm. Finally, in addition to low altitude observations, short pulse data tends to valid nearer cloud edges than long pulse. Manually adjusting weighing functions near clouds to leverage this benefit would be a fairly complex task as the transition regions would likely be quite variable. This makes it impractical to merge long and short pulse data in and around clouds and instead accept lower data availability.

In this work, we demonstrate how advanced processing methods like Poisson Total Variation (PTV), which use forward models, are well suited to the problem of retrieving parameters that are over-constrained and can provide a high quality estimate of products by seamlessly integrating all of the observation channels. This is done in a way that balances the uncertainties of each observation so that the data product is most responsive to the observations that have the lowest uncertainty based on the detector noise model. It also provides continuity in the estimated fields where data from one pulse configuration may not be accurate or available (e.g. low altitudes and near cloud edges).

Line 97-98. This sentence is a little confusing as you reference a detector channel being common across all four channels. Please clarify what channels.

Before Eq. 1, we have added text to try to clarify that different "channels" are created by changing the transmit wavelength and pulse length.

The general water vapor DIAL signal model used here is similar to that employed in \citep{Marais2022} where now the laser pulse may be varied to create different channels for combinations of transmitted wavelength and pulse width.

We have attempted to clarify the text in question by avoiding using the word "channel" in reference to the detector:

Note that because the background is a function of receiver behavior and the same detector is used to acquire optical signals from all four channels, it should be channel independent in the WV DIAL.

Lines 148-150. This is generic statement that should be further clarified or removed without references. DIAL profiling within liquid cloud is clearly a challenge as the extinction of most liquid clouds are sufficiently high such that signals attenuate to the measurement limit well below the resolution of typical DIAL retrievals (~100-300 m). That said, so long as the assumptions of the DIAL equation are not broken (the humidity and cross sections stays relatively constant within a range bin, correcting for Rayleigh-Doppler effects, and signals stay linear and constant between on and off signals), profiling within and near clouds is feasible. This in practice is easier with ice cloud as the extinction is much lower. The DLR WALES airborne DIAL has demonstrated this repeatedly and preliminary cases with the NASA HALO airborne DIAL have also been published. The comment about integrating over heterogneous backscatter is noted. There should be clarifying text to clarify the difference between deficiencies in lidar sampling that would result in errors, vs adequate sampling strategy (fast and high resolution) but averaging over multiple shots to get adequate SNR for a good DIAL or cloud property retrieval. An average value (and variance if available) is still a very scientifically useful value.

Wendisch, M., Crewell, S., Ehrlich, A., Herber, A., Kirbus, B., Lüpkes, C., Mech, M., Abel, S. J., Akansu, E. F., Ament, F., Aubry, C., Becker, S., Borrmann, S., Bozem, H., Brückner, M., Clemen, H.-C., Dahlke, S., Dekoutsidis, G., Delanoë, J., De La Torre Castro, E., Dorff, H., Dupuy, R., Eppers, O., Ewald, F., George, G., Gorodetskaya, I. V., Grawe, S., Groß, S., Hartmann, J., Henning, S., Hirsch, L., Jäkel, E., Joppe, P., Jourdan, O., Jurányi, Z., Karalis, M., Kellermann, M., Klingebiel, M., Lonardi, M., Lucke, J., Luebke, A. E., Maahn, M., Maherndl, N., Maturilli, M., Mayer, B., Mayer, J., Mertes, S., Michaelis, J., Michalkov, M., Mioche, G., Moser, M., Müller, H., Neggers, R., Ori, D., Paul, D., Paulus, F. M., Pilz, C., Pithan, F., Pöhlker, M., Pörtge, V., Ringel, M., Risse, N., Roberts, G. C., Rosenburg, S., Röttenbacher, J., Rückert, J., Schäfer, M., Schaefer, J., Schemann, V., Schirmacher, I., Schmidt, J., Schmidt, S., Schneider, J., Schnitt, S., Schwarz, A., Siebert, H., Sodemann, H., Sperzel, T., Spreen, G., Stevens, B., Stratmann, F., Svensson, G., Tatzelt, C., Tuch, T., Vihma, T., Voigt, C., Volkmer, L., Walbröl, A., Weber, A., Wehner, B., Wetzel, B., Wirth, M., and Zinner, T.: Overview: quasi-Lagrangian observations of Arctic air mass transformations – introduction and initial results of the HALO-(A C)3 aircraft campaign, Atmos. Chem. Phys., 24, 8865-8892, https://doi.org/10.5194/acp-24-8865-2024, 2024.

Groß, S., Wirth, M., Schäfler, A., Fix, A., Kaufmann, S., and Voigt, C.: Potential of airborne lidar measurements for cirrus cloud studies, Atmos. Meas. Tech., 7, 2745–2755, https://doi.org/10.5194/amt-7-2745-2014, 2014.

Dekoutsidis, G., Wirth, M., and Groß, S.: The effects of warm-air intrusions in the high Arctic on cirrus clouds, Atmos. Chem. Phys., 24, 5971–5987, https://doi.org/10.5194/acp-24-5971-2024, 2024.

Nowottnick, E. P., and Coauthors, 2024: Dust, Convection, Winds, and Waves: The 2022 NASA CPEX-CV Campaign. Bull. Amer. Meteor. Soc., 105, E2097–E2125, <a href="https://doi.org/10.1175/BAMS-D-23-0201.1">https://doi.org/10.1175/BAMS-D-23-0201.1</a>.

We thank the reviewer for pointing out that we were overly general in our description of errors related to clouds in DIAL and we have revised the text accordingly to account for the fact that DIAL likely can work in *some* cloud conditions.

It is well known that DIAL estimates of absolute humidity can be nonphysical within cloud structures. These nonphysical values produced in many clouds are typically highly variable, ranging from negative to unrealistically high values.

We want to also caution against over assertion of capability for DIAL in clouds. The citations provided by the reviewer all rely on the Groß et al. work to justify the claim that cirrus retrievals are accurate. That analysis was applied to a single cloud with 1 hour of validation data. Within that work, the justification for errors is entirely explained as the product of inhomogeneity causing differences in sampling between online and offline channels. We don't think this is a comprehensive explanation of the contributing factors (we provide other factors that may contribute errors in the same section). For one thing, MPD is stationary and the online and offline signals are interleaved on a shot-to-shot basis and we have looked at this in considerable detail (see the 2023 AMS talk referenced).

It needs to be recognized that DIAL retrievals are probably sufficiently accurate in *some* clouds, but we also know there are cases where they clearly fail. Knowing where a particular observation lies on that spectrum without a validation reference does not appear to be possible (unless the retrievals are obviously non-physical). That presents a significant barrier to providing observed water vapor from such lidar systems. A comprehensive validation analysis would need to include more than a successful observation, but also demonstrate an ability to correctly predict the errors in inaccurate regions as well.

Line 215. What decrease the histogram time to 2Hz or 4 Hz to capture variability then average further in post-processing?

At present data is captured at 2 second resolution and processed at 5 minute resolution, so this is done, but at a different cadence than suggested by the reviewer. Higher temporal and range resolution is preferable for variability analysis, but a practical trade has been made with regard to data storage and transfer. The MPD operates continuously and typically transfers data off the instrument to data servers using a cell modem. Higher resolution raw data requires more expensive data plans and more data storage on servers.

Figures 1 and 3. Is it possible to show where the contributions are from each 'pulse'? It would be informative to the reader to explicitly see how these get stitched together rather than trying to squint between the panels.

The signal estimates are obtained by optimally fitting all observed channels. Determining how each channel contributes to the ultimate product by minimizing Eq. (3) is not straightforward

(though probably not impossible). However the retrieval includes the smearing effect of the pulses and deconvolves the product. For this reason, a single observation will contribute in varying amounts to multiple pixels in the retrieved variables. Therefore the contributions would be 3D tensors for each channel. Our feeling is that developing and deploying this capability to assess relative contributions represents an excessive effort for the potential benefit to the published work. While we can understand the motivation for this comment, we don't believe there is a straight-forward way to address it. No changes have been made in the manuscript to address it.

Line 310 - remove 'where' between 500 m and due.

**Done**

Figure 5. Given the focus here is on improving the near surface retrievals and PTV has been demonstrated in previous publications to improve performance aloft, it would be ideal to zoom in on the retrievals down near the surface (below 2 or even 1 km) to better elucidate the improvements. As they stand some of the plots are busy and discerning the different lines down near the surface is difficult. It would be good to show a vertical resolution curve for each combined PTV retrieval (on the right y axis?) to give the reader a sense of how DIAL resolution stacks up against the short and long pulse duration.

We feel it is important to show the full profile up to 6 km to contrast the impact of the different pulse combinations (short pulse only does not capture signals as high) and demonstrate the overall benefit of leveraging both pulse configurations. However we do agree with the next comment in this same line of thinking.

Regarding resolutions, these are not explicitly set in the signal processing as PTV will inherently trade resolution for signal based on correlated features in the image. As a result, there is not presently a clear definition for resolution in this processing scheme.

To clarify this, we have added the following text to the section describing PTV: The result is a retrieved data product in which averaging occurs across patches of correlated structure so that the resolution varies across the 2D image and is not constant across the time or range axis.

Table 1. It would be good to compile the statistics for low altitudes (below 1km?) and all altitudes. This way the reader gains a better understanding of the improvements near the surface relative to the overall improvement.

Thank you for the suggestion. We have added the same metrics for altitudes less than 1 km to Table 1 and included some additional discussion of the results.

Paragraph starting at line 349 – Great discussion on the need for future trades.

Thank you!

---

## Author Response (AR2)

Dear authors,

Thank you for responding thoroughly to the reviewers' comments. After considering your exchange with them, I believe the manuscript is now ready for publication. It should be noted that the authors have correctly left some questions open, highlighting possible future developments.

However, I ask the authors to make only one change to a specific point raised by reviewer #2, which, in my opinion, has not been given due consideration.
Reviewer #2 notes that "There should be clarifying text to clarify the difference between deficiencies in lidar sampling that would result in errors, vs adequate sampling strategy (fast and high resolution) but averaging over multiple shots to get adequate SNR for a good DIAL or cloud property retrieval."

The authors rightly note that the work of Gros et al. AMT 2014 is central to the focus of the discussion but do not fully engage in the clarification of how to generally treat DIAL under cloudy conditions, as per reviewer request. It would be appreciated if the authors could amend the relevant paragraph in section 2.2 with a more comprehensive discussion, following the reviewer's suggestion.

Best regards
Luca Lelli

From the second reviewer:
*The comment about integrating over heterogneous backscatter is noted. There should be clarifying text to clarify the difference between deficiencies in lidar sampling that would result in errors, vs adequate sampling strategy (fast and high resolution) but averaging over multiple shots to get adequate SNR for a good DIAL or cloud property retrieval. An average value (and variance if available) is still a very scientifically useful value.*

From the original paper:
*It is well known that DIAL estimates of absolute humidity tend to be nonphysical within cloud structures. The values produced in clouds are typically highly variable, ranging from negative to unrealistically high values. While no comprehensive explanation of this phenomenon has been robustly demonstrated, some of our research on this subject suggests the errors in clouds are combinations of nonlinear observations (both hardware driven due to detector behavior and the nonlinearity of the observation to the water vapor), integrating over heterogeneous backscatter structure (in range and time) and ignoring the effects of the laser pulse convolution (e.g. see Hayman et al. (2023)). The reader should note that, while such errors are obvious in DIAL, this very likely indicates that all lidar retrievals in clouds will have significant unaccounted for errors.*

Response:

Thank you for the feedback. The intent of the Masking section (the section containing the text referenced above) is to provide transparency about how data quality performed in the analysis presented in this paper. We agree that investigating sampling strategy is an interesting and potentially important research topic, but would likely need its own dedicated publication. We don't feel we can accurately comment on this without significantly deviating from the theme of this work. In an effort to clarify the purpose of the section and provide a better context for the masking typically conducted in DIAL analysis, we have substantially modified the opening of the Masking section. This includes more discussion about the possibility of clouds where data may be valid and a reference to Groß et al 2014. This new text is printed below for your convenience.

*It is common practice to mask cloud structure in DIAL estimates due to the frequent occurrence of biases in retrieved water vapor concentration. These biases are often easy to identify because the retrieved quantities are non-physical (e.g. negative water vapor or unrealistically large values) in the cloud structures. There is relatively little published research on the root cause of data quality issues for DIAL in cloud, and the dominant sources of issues may vary between hardware architectures. Differences in online and offline sample volumes, laser pulse length convolution, detector nonlinearity, and Rayleigh-Doppler effect are common explanations for the error in most clouds; however, our own efforts to demonstrate and isolate the impact of these and other errors have been difficult to comprehensively validate experimentally. It seems entirely possible that there are additional contributing factors. It is also likely that there are some conditions where DIAL retrievals are valid in certain types of clouds and under certain conditions. For example, \cite{Gros2014} showed DIAL retrievals in high altitude cirrus that agreed well with an in situ sensor on a second aircraft over a 30 minute period. However, the analysis presented in that work was fairly limited, making it difficult to generalize even to other cirrus cloud cases. For now, it is not entirely clear what all the interacting conditions are that influence DIAL retrieval accuracy or if it is possible to know if those conditions are satisfied from the lidar data alone (meaning no external validation source would be required). For this reason it is often generalized that DIAL data is valid in clear air and most aerosol loaded conditions, but not in clouds.*

*Our masking routine developed here is based on the hypothesis that much of the bias seen in DIAL observations of clouds originates from averaging over heterogeneous backscatter structure \citep{HaymanAMS2023}\footnote{the presentation is available at minute 32 here https://ams.confex.com/ams/103ANNUAL/meetingapp.cgi/Session/63551}. However, based on ultra-high-resolution observations of clouds by MPD (similar to that reported in \cite{HaymanSciRep2024} captured at 8 kHz and 5 ns resolution and processed to 50 Hz and 75 cm resolution) we do not believe that this represents a \emph{fully} comprehensive description of why most clouds produce non-physical water vapor retrievals. Nevertheless, masking data that produce errors is a necessary component of DIAL processing. The masking approach described here represents a heuristic approach based on the described reasoning. Although masking would likely be further improved if more comprehensive understanding of*

*these errors were robustly demonstrated, the approach outlined here represents an improvement over the methods we had described previously.*